# Immunogenicity and Safety of a Booster Dose of Live Attenuated Varicella Vaccine, and Immune Persistence of a Primary Dose for Children Aged 2 to 6 Years

**DOI:** 10.3390/vaccines10050660

**Published:** 2022-04-22

**Authors:** Yukai Zhang, Lei Wang, Yanxia Wang, Wei Zhang, Ningning Jia, Zhiqiang Xie, Lili Huang, Wangyang You, Weifeng Lu, Erwei Li, Feilong Gao, Yuansheng Hu, Fanhong Meng, Shengli Xia

**Affiliations:** 1Clinical Laboratory, Jinshui District Center for Disease Control and Prevention, Zhengzhou 450003, China; kk13937184288@126.com; 2Clinical R&D Center, Sinovac Biotech Co., Ltd., Beijing 100085, China; wanglei1@sinovac.com (L.W.); jiann@sinovac.com (N.J.); 3Vaccine Clinical Research Center, Henan Provincial Center for Disease Control and Prevention, Zhengzhou 450018, China; wangyanxia99@163.com (Y.W.); zwzzu@163.com (W.Z.); xiezqshang@163.com (Z.X.); 13643826177@163.com (L.H.); dsrt12345@163.com (W.Y.); 4Xiangfu District Center for Disease Control and Prevention, Kaifeng 475199, China; kfxcdclwf@163.com (W.L.); lew998@126.com (E.L.); xfqcdcgfl@163.com (F.G.); 5Research Department, Sinovac (Dalian) Vaccine Technology Co., Ltd., Dalian 116620, China

**Keywords:** live attenuated varicella vaccine, booster, immune persistence, immunogenicity, safety

## Abstract

Aim: To evaluate the immunogenicity and safety of a booster dose of live attenuated varicella vaccine (VarV) manufactured by Sinovac (Dalian) Vaccine Technology Co. Ltd., and the immune persistence of a primary dose in 2- to 6-year-old children. Methods: A phase IV, open-label study was conducted in China. Children previously vaccinated with a single dose of VarV at 1~3 years old received one dose of homologous VarV in the first year, the second year, or the third year after the primary immunization as booster immunization. Immune persistence was evaluated in an immune persistence analysis set, while immunogenicity was evaluated in a per-protocol analysis set, and safety was evaluated in a safety analysis set. The primary endpoint was the seropositive rate and the seroconversion rate of VarV antibody. The trial was registered at ClinicalTrials.gov (NCT02981836). Results: From July 2018 to August 2020, a total of 849 vaccinated children received the booster vaccination of VarV, one booster dose for each child (301 vaccinated in the first year after primary immunization (Group 1), 276 vaccinated in the second year after primary immunization (Group 2), 272 vaccinated in the third year after primary immunization (Group 3)). The seropositive rates were 99.34%, 97.83%, and 98.16% in Groups 1–3, with GMTs of 1:22.56, 1:18.49, and 1:18.45, respectively. Thirty days after the vaccine booster dose, the seropositive rates of the three groups were all 100% and the seroconversion rates were 52.54%, 67.46%, and 66.67%, with GMTs of 1:68.49, 1:76.32 and 1:78.34, respectively. The seroconversion rates in Groups 2 and 3 were both higher than that in Group 1 (*p* = 0.0005 and *p* = 0.0008). The overall incidence of adverse reactions was 7.77%, with 7.64%, 8.33%, and 7.35% in Groups 1, 2, and 3, respectively. The main symptom among adverse reactions was fever, the incidence of which ranged from 5.07% to 6.64% in each group, and no vaccine-related serious adverse events occurred. Conclusions: VarV had good immune persistence in 1~3 years after primary immunization. A vaccine booster dose for children aged 1~3 years after primary immunization recalled specific immune response to varicella-zoster virus, with no safety concerns increased.

## 1. Background

Varicella-zoster virus (VZV) is an alpha herpes virus belonging to the Herpesviridae family [1] and a pathogenic human alpha-herpesvirus that causes chickenpox (varicella) as a primary infection, which usually occurs in children in locales where vaccination is not practiced [2]. Varicella is a worldwide and airborne disease spread by coughing and sneezing, and also by contact with skin lesions [3]. VZV activates, either spontaneously or following one or more of various triggering factors, to cause herpes zoster (shingles), which usually appears as a painful or pruritic cutaneous vesicular eruption distributed in characteristic dermatomal [2,4]. The secondary household attack rate of over 90% shows that varicella is highly contagious [5]. Varicella vaccination has been proven to be effective in seroconverting pediatric patients (including children with lukaemia), adolescents and adults, with a low occurrence of vaccine-associated rash among immunocompetent patients [6]. The varicella vaccination can prevent about 70~90% of infections and 95% of severe diseases [3].

Though the universal varicella vaccination has led to a reduction in morbidity and mortality resulting from varicella infection in the USA, active surveillance data from sites and states with well-implemented vaccination programs indicate that the number of reported varicella cases remains constant or has declined minimally [7]. However, outbreaks of varicella in schools with high vaccine coverage rate are often reported [8]. Considering that a single dose has been shown to be effective in reducing the incidence of severe varicella, this schedule has been associated with breakthrough disease [9,10] caused by primary vaccine failure and the declining exogenous exposure from children shedding wild-strain VZV [11]. As a result, a two-dose schedule is recommended for optimal protection against varicella of any severity and to prevent risks of breakthrough varicella and outbreaks [12]. Further, a global meta-analysis of 42 studies estimates that a single dose of varicella vaccine is moderately effective in preventing all varicella (81%) and highly effective at preventing combined moderate and severe varicella (98%), while two doses of varicella vaccine are highly effective at preventing all varicella [13].

Live attenuated varicella vaccine (VarV) is an attenuated strain of VZV (Oka strain) that is used to inoculate human diploid cells (SV-1 strain), which is made by culturing and harvesting the virus, adding suitable stabilizers and freeze-drying. After vaccination, it can stimulate the body to produce immunity against the varicella-zoster virus, therefore, VarV is used to prevent varicella. VarV has shown good safety and immunogenicity in the phases I–III studies. However, there are few trials about the immunogenicity and safety of a booster dose of VarV in children. Hence, on 11 July 2018, the Henan Provincial Center for Disease Control and Prevention was commissioned to carry out a phase IV clinical trial to evaluate the immunogenicity and safety of a booster dose of VarV, and immune persistence of a primary dose in 2- to 6-year-old children.

## 2. Methods

### 2.1. Study Design

The Henan Provincial Center for Disease Control and Prevention (HNCDC) is in charge of the phase IV, open-label study, which was performed in Xiangcheng County, Henan Province, China. In this study, the HNCDC was responsible for collecting data and conducting data analysis. In this trial, healthy young children aged 1 to 3 years were recruited from July 2018 to August 2020. The protocol for the clinical trial and the informed consent form were reviewed by the Ethics Committee of the HNCDC (2016-YM-001-02). In addition, the clinical trial was registered at ClinicalTrials.gov (NCT02981836).

### 2.2. Participants

Subjects were recruited at the Xiangfu District Center for Disease Control and Prevention. Inclusion criteria of subjects: (1) participated in three batches of consistent clinical trials in phase III and received a dose of VarV; (2) informed consent of the subject’s guardian obtained and the guardian signed the informed consent form. Exclusion criteria of subjects: (1) vaccinated with VarV after the end of three batches of consistent clinical trials in phase III; (2) could not participate in the clinical trial according to the judgement of the investigators.

### 2.3. VarV Details

Human diploid cells (SV-1 strain) were inoculated with the attenuated varicella-zoster virus strain (Oka strain). The virus was cultured, harvested, and lyophilized by adding appropriate stabilizer. VarV was a white loose body, which turned to a clear liquid after redissolution, with slight opalescence. After redissolution, the liquid in each vial was 0.5 mL, which was for each human dose. The investigational product was a colorless clear liquid with a specification of 0.5 mL/piece. VarV contained no less than 3.3 lg PFU. It is qualified by the China National Institutes for Food and Drug Control. Vaccine diluent was sterile water for injection, produced by the Jiangsu Desenuo Pharmaceutical Co., Ltd., Liyang, China. It was qualified by the manufacturer and the Sinovac (Dalian) Vaccine Technology Co. Ltd, Dalian, China. VarV was stored and transported in the dark at 2~8 °C, and freezing was strictly prohibited.

### 2.4. Procedures

A total of 1195 subjects were enrolled. They were vaccinated with one initial single dose of VarV in three batches of consistent clinical trials in phase III, and then divided into three groups based on their study number, namely C0001—C0400, C0401—C0800, and C0801—C1197. Among them, C0243 and C0556 were excluded because they were not vaccinated in the three batches of consistent trials. They received subcutaneous injections of 0.5 mL VarV into the lower edge of the lateral deltoid muscle each time. The investigators observed the seropositive rate of serum antibody, serum antibody GMT, and adverse effects after the booster vaccination. According to the time of booster immunization, they were divided into three groups. In the first group (Group 1), subjects were vaccinated in the first year after primary immunization; in the second group (Group 2), subjects were vaccinated in the second year after primary immunization; in the third group (Group 3), subjects were vaccinated in the third year after primary immunization. The seropositive was defined as VZV antibody titer ≥1:4. About 3.0 mL of venous blood was collected from all subjects before booster immunization and 30 days after booster immunization for antibody detection. The serum samples were sent simultaneously and tested in blind state using the fluorescent antibody to membrane antigen (FAMA) method, which has become a preferred method to determine the serodiagnosis of VZN infection. There were no testing kits used in the process. The fluorescent detection plates were prepared with Oka virus-diploid cell suspensions and stored at −70 °C. Serum samples were separated from blood samples, and inactivated in a water bath at 56 °C for 30 min and stored at −20 °C. Diluted in 2-fold serials, samples were dropped into fluorescent plate wells with positive and negative controls. The results were observed under fluorescence microscopes after incubation with test antibodies and Evan Blue. In the samples to be tested that were positive, varicella antibody was bound to the surface antigens of positive cells infected with the varicella virus, and after the binding to the fluorescent secondary antibody, a bright green halo appeared under the fluorescence microscope. If no green halo and only red cells were observed, it was judged as negative; if the serum dilution ratio was less than 1:4, it was also judged as negative.

### 2.5. Outcomes

The endpoints of the trial included the primary endpoint and the secondary endpoint. The primary endpoint included: the seropositive rate of serum antibody in Groups 1–3, the seroconversion rate of serum antibody 30 days after booster immunization, and the seropositive rate of serum antibody 30 days after booster immunization. The secondary endpoint included: serum antibody geometric mean titer (GMT)/geometric mean increase (GMI) before booster immunization and 30 days after booster immunization, and the incidence of local and systemic adverse events (AE), adverse reactions, and serious adverse events (SAE) after booster immunization.

### 2.6. Statistical Analysis

Qualitative indicators such as seropositive rate, seroconversion rate, and incidence of adverse reactions are listed in the frequency distribution table, and quantitative indicators such as antibody titer/concentration are reported in the form of the mean ± standard deviation, median, maximum and minimum, and 95% confidence interval. Quantitative indicators were tested for normality at first. According to the distribution characteristics of variables, the ANOVA/Wilcoxon rank sum test were used to statistically compare the differences of age, height, weight, etc. among groups. The chi-square test/Fisher’s exact probability test were used to statistically compare the differences of sex ratio among groups (Table 1); the chi-square test/Fisher’s exact probability test were used to statistically compare the differences of positive rate (≥1:4) at different time points in each group. ANOVA/Wilcoxon rank sum test were used to statistically compare the differences of GMT at different time points in each group (Table 2); the chi-square test/Fisher’s exact probability test were used to statistically compare the differences of positive rate (≥1:4) among groups. The ANOVA/Wilcoxon rank sum test were used to statistically compare the differences of GMT and GMI among groups (Table 3). The *p*-value was calculated by Fisher’s exact probability method (Table 4), and *p* ≤ 0.05 was considered to be statistically significant in the test. All statistical tests were given the results of test statistics and *p*-values, and all statistical analyses were carried out with SAS 9.4 (SAS Institute, Cary, NC, USA).

## 3. Results

### 3.1. Study Population

A total of 849 subjects were enrolled in this study, including 301 subjects in Group 1, 276 subjects in Group 2, and 272 subjects in Group 3; all subjects completed the study without dropping out, with all of them included in the safety analysis set (SS), the full analysis set (FAS), and the immune persistence set (IPS). A total of 792 subjects were included in per protocol set (PPS), including 276 subjects in Group 1, 252 subjects in Group 2, and 264 subjects in Group 3. The mean age was 45.23 months in Group 1, 61.47 months in Group 2, and 70.4 months in Group 3. In addition, other characteristic information of subjects is shown in Table 1.

### 3.2. Immune Persistence

Thirty days after primary immunization, the seropositive rates of varicella antibody (≥1:4) in Groups 1–3 were 97.67%, 97.10%, and 98.90%, respectively, and GMTs were 1:29.05, 1:24.40, and 1:22.40, respectively. In 1–3 years after vaccination, the seropositive rates of varicella antibody (≥1:4) in Groups 1–3 were 99.34%, 97.83%, and 98.16%, respectively. The seropositive rates of 30 days after basic immunization were compared to those of 1–3 years after vaccination, and the results showed that the differences were not statistically significant. GMTs of 1~3 years after vaccination were 1:22.56, 1:18.49, and 1:18.45, respectively, showing a decreasing trend over time and lower than those of 30 days after basic immunization, and the differences were statistically significant (*p* = 0.0002, *p* = 0.0003, and *p* = 0.0040). More details are shown in the Table 2.

### 3.3. Booster Immunogenicity

Thirty days after booster immunization, the seropositive rates of varicella antibody (≥1:4) were all 100% in Groups 1–3, and the seroconversion rates of varicella antibody (≥1:4) in Groups 1–3 were 52.54%, 67.46% and 66.67%, respectively. The seroconversion rates in Groups 2 and 3 were higher than that in Group 1, and the differences were statistically significant (*p* = 0.0005 and *p* = 0.0008). GMTs were 1:68.49, 1:76.32, and 1:78.34, respectively, and the difference was not statistically significant (*p* = 0.2663). GMIs were 3.06, 4.13, and 4.22, respectively. GMIs of Groups 2 and 3 were both higher than that of Group 1, and the differences were statistically significant (*p* = 0.0002 and *p* = 0.0001). More details are shown in the Table 3.

### 3.4. Safety of Booster Immunization

AEs occurred in 9.19% (78/849) of subjects after vaccination. The incidences of AEs were 8.97% (27/301), 10.14% (28/276), and 8.46% (23/272), in Groups 1–3, respectively, and the difference was not statistically significant (*p* = 0.7746). After vaccination, 7.77% (66/849) of subjects had adverse reactions. The incidences of adverse reactions were 7.64% (23/301), 8.33% (23/276), and 7.35% (20/272), in Groups 1–3, respectively, and the difference was not statistically significant (*p* = 0.9102). Adverse reactions occurred within 0–14 days after booster immunization. The incidences of adverse reactions within 30 min were 0.33% (1/301), 0.36% (1/276), and 2.57% (7/272), respectively, in Groups 1–3. The incidence of adverse reactions in the group 3 was higher than those of Groups 1 and 2, and the difference was statistically significant (*p* = 0.0166). Moreover, the results of solicited local AEs (pain, erythema (redness), pruritus, and induration/swelling at the injection site), solicited systemic AEs (diarrhea, nausea/vomiting, cough, myalgia (non-inoculated site), allergic reaction, headache, fatigue and fever), and other information are shown in the Table 4.

After vaccination, the adverse reactions of 849 subjects were mainly solicited reactions. The incidences of adverse reactions were 7.64%, 8.33%, and 7.35%, in Groups 1–3, respectively. The main symptom was fever, the incidences of fever were 6.64%, 5.07%, and 6.25%, in Groups 1–3, respectively, and the difference was not statistically significant (*p* = 0.7232). The incidence of other adverse reactions and symptoms in each group were basically <0.01.1%, except cough (*p* = 0.0174), and the differences were not statistically significant among all groups. During the trial, only one SAE occurred in Group 2 after vaccination. A 44-year-old man was hospitalized 20 days after the vaccination due to fever and diarrhea. The patient’s symptoms lasted for 6 days, and then disappeared after the treatment. According to the doctor’s diagnosis, the occurrence of this symptom was not related to the vaccination. The incidence of SAE was 0.36% (1/276), and there was no statistically significant difference in the incidence of SAE among groups (*p* = 0.6455). No vaccine-related SAE occurred, and the SAE was enteritis with upper respiratory tract infection.

## 4. Discussion

Reaching the high vaccine coverage rates, which is necessary to achieve the full benefits of varicella vaccination in children, would be facilitated by the existing two-dose vaccination [14]. Therefore, VarV can be used to increase two-dose varicella vaccine coverage. More importantly, the results showed that the booster vaccination of VarV has good immune persistence. Based on the immunological results of the trial, the seropositive rates of varicella antibody (≥1:4), 30 days after the first dose of VarV, in three groups, were 97.67%, 97.10%, and 98.90%, and GMTs were 1:29.05, 1:24.40, and 1:22.40, respectively. In a study by LIU Ye et al. [15], freeze-dried live attenuated varicella vaccine was used to observe the immunization effect in children for 5 years, and the seropositive rates of varicella antibody (≥1:4) were 91.05%, 89.39%, and 95.56%, respectively. These values are slightly lower than the results of this study. The reason is that in the study by LIU Ye et al., they only observed the immune persistence of one dose of VarV to indicate the importance of booster vaccination of varicella vaccine. In another study [16], children aged 1–7 years were given a booster dose of VarV in the first year, the third year, and the fifth year after the initial vaccination, and the results showed that the seropositive rates (≥1:4) were 93.10% and 50.00% in Groups 1 and 3, respectively, and GMTs were 1:12.56 and 1:6.81, respectively. The seropositive rates and GMTs are lower than the results of this study, which further confirms that VarV has good immune persistence and better immune protection. The seropositive rates of antibody (≥1:4) in this study were all 100% in Groups 1–3, higher than that of Mitra’s two doses of VarV (98.3%) [17]. This also shows the superiority of the VarV developed by the Sinovac (Dalian) Vaccine Technology Co. Ltd.

Some researchers suspect that a second dose given too early after the previous dose may reduce the response to that dose [7,18]. Based on this, it was also found in this study that the level of antibody after immunization was basically the same as that of Groups 2 and 3, and was better than that of Group 1 before the second dose. This may be related to the high antibody level in Group 1 after immunization. In a study by Liu, S.K. et al. [16], it was shown that the seroconversion rates of varicella antibody were 100% in Groups 1 and 3, and the GMT was 1:84.82 in Group 1, both higher than the results of the present study, as well as 1:40.65 in Group 3, lower than the results of the present study. Considering that it may be related to the difference in the levels of pre-immunization antibodies in different studies, it is suggested that the time interval of VarV booster immunization should not be too short, and children should receive a dose of booster immunization in the second or third year after basic immunization.

VZV vaccines are generally safe and well tolerated, and the reported adverse events related to all VZV vaccines are usually either mild or moderate in nature. The Global Safety Committee listed herpes zoster, pain, and rash as the three most frequent AEs for VZV vaccine in their report in 2013 [17]. In our study, after 849 subjects were vaccinated, the overall incidence of adverse reactions was 7.77%, and the incidence of adverse reactions in Groups 1 to 3 were 7.64%, 8.33%, and 7.35%, respectively. The main symptom of adverse reactions was fever, and the incidences of adverse reactions were 5.07~6.64% among the three groups. However, the incidence of adverse reactions within 30 min was significantly higher in Group 3. The details of adverse reactions within 30 min were fever (five persons), diarrhea (one person) and headache (one person). The results of AEs and adverse reactions are higher than those of the study by John R. Su [19]. The reason is that he analyzed a total of 14,641 subjects from the Vaccine Adverse Event Reporting System from 2006 to 2014 among children aged 4 to 18 years, whereas our sample size was relatively small. The sample size of this study was calculated with reference to the results of previous clinical studies. The positive rate of varicella antibody 30 days after booster immunization was the observation index, which was more than 90% in references. The sample size was calculated according to 90% conservatively, taking α = 0.05 (bilateral) and 95% credible interval. The difference between the upper limit and the lower limit of the interval was taken as 12%, which was calculated using the NCSS-PASS software, and the number of samples was 112. Considering that there may be some loss to follow-up or exclusion during the trial, a conservative estimate was made based on the rate of loss to follow-up of 20%, and the minimum sample size per year was 140. The actual enrolment sample size for each group was higher than the originally assumed minimum sample size. Therefore, the sample size of our study is sufficient for this trial. As compared with other available varicella vaccines, vaccines produced by the Sinovac (Dalian) Vaccine Technology Co. Ltd. adopt a new human diploid cell (SV-1 cell) to culture Oka strain. The sources and donors of SV-1 cells are clear, which is not only in compliance with the legal and ethical requirements, but also meets the requirements of the Chinese Pharmacopoeia, ICH, WHO and FDA. Moreover, the acquisition of the strain does not introduce chemicals, improving the safety of the vaccine. Furthermore, no SAE related to the vaccine occurred. Overall, VarV was well tolerated throughout the study period, indicating that VarV is safe.

## 5. Conclusions

At present, there are several marketed varicella vaccine products in China. However, the market demand for VZV remains far from saturation and the need for VZV is ever-increasing after the inclusion of VZV in the National Immunization Program management system [20]. The existing studies about VZV have mainly focused on its safety, efficacy, or immunogenicity. However, there are limited studies about a booster dose of VZV, which is also a significant indicator to evaluate the efficacy and safety of VarV. This study shows that a booster vaccination of VarV for children aged 2 to 6 years has good immune persistence and safety. It is recommended that children aged 1 to 3 years receive booster shots in the second or third year after the initial vaccination.

## Figures and Tables

**Table 1 vaccines-10-00660-t001:** Demographic characteristics of subjects in different groups.

Analysis Sets	Indicators	Group 1 *	Group 2 #	Group 3 ^¶^	Total	*p*
FAS/IPS	N	301	276	272	849	-
	Age at booster vaccination (months)	45.23 ± 10.55	61.47 ± 9.99	70.04 ± 10.49	58.46 ± 14.66	<0.0001
	Sex ratio (male/female)	1.23	1.11	1.21	1.18	0.7970
	Height (cm)	100.60 ± 7.73	110.05 ± 7.24	115.53 ± 8.08	108.45 ± 9.89	<0.0001
	Weight (kg)	15.88 ± 2.65	19.76 ± 3.12	21.04 ± 4.45	18.80 ± 4.11	<0.0001
PPS	N	276	252	264	792	
	Age at booster vaccination (months)	45.13 ± 10.57	61.38 ± 10.05	69.84 ± 10.52	58.54 ± 14.69	<0.0001
	Sex ratio (male/female)	1.28	1.1	1.2	1.19	0.6836
	Height (cm)	100.42 ± 7.75	110.04 ± 7.36	115.41 ± 8.13	108.48 ± 9.98	<0.0001
	Weight (kg)	15.79 ± 2.66	19.74 ± 3.16	21.02 ± 4.48	18.79 ± 4.17	<0.0001

* Group 1, the first group was subjects vaccinated in the first year after primary immunization; # Group 2, the second group was subjects vaccinated in the second years after primary immunization; **^¶^** Group 3, the third group was subjects vaccinated in the third years after primary immunization.

**Table 2 vaccines-10-00660-t002:** Comparison of antibody levels between subjects at different time points after vaccination and 30 days after basic immunization.

Indicators	Group 1			Group 2			Group 3		
	30 Days *	1st Year *	*p*	30 Days *	2nd Year *	*p*	30 Days *	3rd Year *	*p*
N	301	301		276	276		272	272	
Seropositive rate (≥1:4) n (%)	294 (97.67)	299 (99.34)	0.1764	268 (97.10)	270 (97.83)	0.7878	269 (98.90)	267 (98.16)	0.7245
(95% CI)	(95.27, 99.06)	(97.62, 99.92)		(94.37, 98.74)	(95.33, 99.20)		(96.81, 99.77)	(95.76, 99.40)	
GMT	29.05	22.56	0.0002	24.40	18.49	0.0003	22.40	18.45	0.0040
(95% CI)	(26.27, 32.13)	(20.72, 24.56)		(22.01, 27.05)	(16.57, 20.62)		(20.45, 24.53)	(16.76, 20.31)	

* indicate the number of days after basic immunization.

**Table 3 vaccines-10-00660-t003:** The results of booster immunogenicity.

	Indicators	Group 1	Group 2	Group 3	*p*	*p* * 1 vs. 2	*p* # 1 vs. 3	*p*^¶^ 2 vs. 3
	N	276	252	264				
Before booster immunization	Seropositive rate (≥1:4) n (%)	274 (99.28)	246 (97.62)	259 (98.11)	0.2824			
	(95% CI)	(97.41, 99.91)	(94.89, 99.12)	(95.64, 99.38)				
	GMT (1:)	22.41	18.49	18.58	0.0025	0.0025	0.0032	0.8492
	(95% CI)	(20.50, 24.51)	(16.46, 20.77)	(16.85, 20.49)				
30 Days after booster immunization	Seropositive rate (≥1:4) n (%)	276 (100.00)	252 (100.00)	264 (100.00)	1.0000			
	(95% CI)	(98.67, 100.00)	(98.55, 100.00)	(98.61, 100.00)				
	Seroconversion rate (≥1:4) n (%)	145 (52.54)	170 (67.46)	176 (66.67)	0.0003	0.0005	0.0008	0.8480
	(95% CI)	(46.46, 58.55)	(61.30, 73.21)	(60.63, 72.33)				
	GMT (1:)	68.49	76.32	78.34	0.2663			
	(95% CI)	(60.99, 76.91)	(67.28, 86.58)	(68.89, 89.09)				
	GMI	3.06	4.13	4.22	<0.0001	0.0002	0.0001	0.9978
	(95% CI)	(2.77, 3.38)	(3.65, 4.66)	(3.73, 4.77)				

* The *p*-value of Group 1 as compared with that of Group 2; # the *p*-value of Group 1 as compared with that of Group 3; **^¶^** the *p*-value of Group 2 as compared with that of Group 3.

**Table 4 vaccines-10-00660-t004:** The details of adverse events after booster immunization.

Classification	Group 1 (N = 301)		Group 2 (N = 276)		Group 3 (N = 272)		Total (N = 849)		*p*
	n	Incidence (%)	n	Incidence (%)	n	Incidence (%)	n	Incidence (%)	
Overall adverse events	27	8.97	28	10.14	23	8.46	78	9.19	0.7746
Adverse events not related to vaccines	5	1.66	5	1.81	6	2.21	16	1.88	0.9042
Vaccine-related adverse events	23	7.64	23	8.33	20	7.35	66	7.77	0.9102
Solicited adverse events	23	7.64	21	7.61	20	7.35	64	7.54	1.0000
Unsolicited adverse events	0	0.00	2	0.72	0	0.00	2	0.24	0.2078
Within 30 min	1	0.33	1	0.36	7	2.57	9	1.06	0.0166
0~14 Days	23	7.64	23	8.33	20	7.35	66	7.77	0.9102

## Data Availability

Not applicable.

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
