# Peer review of "Immunogenicity and Safety of a Booster Dose of Live Attenuated Varicella Vaccine, and Immune Persistence of a Primary Dose for Children Aged 2 to 6 Years"

_vaccines, 2022, doi:10.3390/vaccines10050660_

Round 1

Reviewer 1 Report

This is a well designed and well performed study focused in an important matter.

There are several issues that must be addressed.

  1. In the Methods section, the technique used to measure the antibodies must be made explicit and explained, or the procedure cited in the references. 
  2. Some initials like GMT and GMI should be defined. The term "conversion rate", probably meaning seroconversion rate, should be defined too.
  3. In Table 3, lower part, in the column "Indicators",  two different rows are equally marked as "Positive rate (>1:4) n(%)" but they show different data. In the same Table, the character "¶" that qualify the last column must be explained, but the character is missing in the bottom lines. 
  4. In Table 4, explain what is the difference between columns marked "cases" and "n". Also, please explain what the terms "Collective"/"Non-collective" mean.
  5. In the text, line 195, it says that there was only one severe adverse event (SAE); apparently it was no related to the vaccine; however I think it must be described.
  6. In Line 96, it says, "VarV was inoculated with human diploid cells...". The meaning of this is not clear; it must be rephrased. A general revision of the English language is warranted. 

Reviewer 2 Report

Zhang et al. reported “Immunogenicity and safety of a booster dose of Live Attenuated Varicella Vaccine, and immune persistence of a primary dose in 2-6 years old children”.  There are few reports on the booster of the varicella vaccine, and this report clarified the more detailed booster effect. Therefore, basically, this paper is suitable for publication. Some suggested improvements are shown below.

1, Major points

Deeper discussions need to be expected.

 1-1, What are the factors that did not reduce the antibody titer compared to Ref. [15]? (Line206-209)

 1-2, Why the adverse events in this study were different from Ref. [16]? (Line224-229)

 1-3, The incidence of adverse reactions within 30 minutes was significantly higher in group 3 (Table 4). Discussion should be necessary.

2, Minor points

 2-1, Abbreviations need to be shown the first time they are used, such as GMT and GMI.

 2-2, What is å’Œ? (L27)

 2-3, In Table 3, is the third “positive rate” correct? Positive conversion rate?

 2-4, Part of the table 4 is not unified, such as ().

Reviewer 3 Report

Estimated Authors of the paper "Immunogenicity and safety of a booster dose of Live Attenuated Varicella Vaccine, and immune persistence of a primary dose in 2-6 years old children"

I've read your article with great interest. In this paper, Author report on their study on around 800 recipient of a live and attenuated Varicella Vaccine from OKA cells. The formulate was tested shortly before the inception of the SARS-CoV-2 pandemic, therefore the present article does not deal with the alleged interference between VZV infection and SARS-CoV-2 infection / vaccination. Focusing on the results hereby reported, Authors suggest that the assessed formulate may be able to achieve good and long-lasting seroconversion rates, particularly after a second dose.

As a consequence, I think that the present paper may deserve a full publication on Vaccines, but some adjustements are required.

First of all, as the definition of the three groups is critical for the proper understanding of the paper, please report a summary recap in the caption of the tables where these groups are reported and discussed.

Second, as the statistical comparisons in the various tables are not the same, please include in the captions of the table what test the reported P deals with.

Third, please discuss in greater details the potential advantages of this formulate against those that are yet commercially available, or which technical improvements have recommended the present research.

Fourth, some glimpses on the potential interference of VZV vaccine on SARS-CoV-2 infection / vaccines may be interesting (please retain the present point as a mere suggestion that you are not strongly recommended to address).

Fifth: you paper reports on less than 1000 cases in groups that are relatively small (around 250 participants each). Please discuss in further details how the size of this sample may affect the eventual reliance on your results (i.e. have you performed a preventive power analysis?)

Finally, please be aware that scattered across the text there are a lot of "typos" associated with the use of non-latin fonts and even some ideograms. Please fix it. 

Reviewer 4 Report

The manuscript represents a follow up for booster vaccination in the pediatric population in China. The article provides evidence of the vaccine boost of the immune response and the adverse effects observed after vaccination. In general, the methodology is standard and the statistics appropriate. There is only a minor issue concerning adverse effects which should be specified in the paper since it is a live attenuated vaccine.

The discussion should be enhanced by comparing the different vaccine schemes and the adverse effects.

There are minor details in the text to be corrected.
